# Bioactive Compounds and In Vitro Antioxidant and Anticoccidial Activities of *Opuntia ficus-indica* Flower Extracts

**DOI:** 10.3390/biomedicines11082173

**Published:** 2023-08-02

**Authors:** Meriem Amrane-Abider, Mirela Imre, Viorel Herman, Nedjima Debbou-Iouknane, Salima Zemouri-Alioui, Souad Khaled, Cilia Bouiche, Cristina Nerín, Ulaș Acaroz, Abdelhanine Ayad

**Affiliations:** 1Centre de Recherche en Technologies Agroalimentaires, Route de Targa Ouzemmour, Campus Universitaire, Bejaia 06000, Algeria; meriem.amrane@crtaa.univ-bejaia.dz (M.A.-A.); celia.bouiche@crtaa.univ-bejaia.dz (C.B.); 2Department of Parasitology and Parasitic Disease, Faculty of Veterinary Medicine, University of Life Sciences “King Mihai I” from Timisoara, 300645 Timisoara, Romania; 3Department of Infectious Diseases and Preventive Medicine, Faculty of Veterinary Medicine, University of Life Sciences “King Mihai I” from Timisoara, 300645 Timisoara, Romania; viorel.herman@fmvt.ro; 4Department of Environment Biological Sciences, Faculty of Nature and Life Sciences, University of Bejaia, Bejaia 06000, Algeria; ecokarima@gmail.com; 5Laboratory of Applied Biochemistry, Faculty of Nature and Life Sciences, University of Bejaia, Bejaia 06000, Algeria; zemourisalima.bio@gmail.com; 6Laboratory of Biomathematics, Biochemistry, Biophysics and Scientometrics, Faculty of Nature and Life Sciences, University of Bejaia, Bejaia 06000, Algeria; kh.souad8@yahoo.fr; 7Aragón Institute for Engineering Research (I3A), University of Zaragoza, Campus Rio Ebro, María de Luna 3, 50018 Zaragoza, Spain; cnerin@unizar.es; 8ACR Bio Food and Biochemistry Research and Development, Afyonkarahisar 03200, Turkey; ulasacaroz@hotmail.com; 9Department of Food Hygiene and Technology, Faculty of Veterinary Medicine, Afyon Kocatepe University, Afyonkarahisar 03200, Turkey; 10Department of Food Hygiene and Technology, Faculty of Veterinary Medicine, Kyrgyz-Turkish Manas University, Bishkek KG-720038, Kyrgyzstan

**Keywords:** *Opuntia ficus-indica* flower, microwave, optimization, antioxidant activity, anticoccidial activity

## Abstract

The objective of the present study is to identify the biochemical compounds extracted from OFI flowers using ultra-high-performance liquid chromatography–electrospray ionization quadrupole time-of-flight mass spectrometry and to evaluate their in vitro antioxidant activities and anticoccidial effects on the destruction of *Eimeria* oocysts isolated from naturally infected chickens. A domestic microwave was used with a refrigerant to condense the vapors generated during the extraction. The flavonoid and phenolic compound contents of the OFI flowers were determined according to standard methods. DPPH radical and H_2_O_2_ scavenging capacities were used to assess the antioxidant activity. Regarding the anticoccidial activity, the *Eimeria* spp. oocysts used were isolated from the fresh feces of infected broilers and were determined in triplicate by incubation at an ambient temperature for 24 h. The results highlighted the considerable influence of the optimized acetone concentration, ratio, irradiation time, and microwave power parameters on the phenolic content and antioxidant activities. Our results revealed significant matches between the predicted and experimental values of the models. Molecular analysis revealed the presence of several biophenol classes such as quercetin, isorhamnetin 3-O-rutinoside, and quercetin-3-O-rutinoside. OFI flower extracts inhibited sporulation and damaged the morphology of *Eimeria* oocysts compared with normal sporulated *Eimeria* oocysts containing sporocysts. In conclusion, the optimized conditions were validated and found to fit very well with the experimental values. These findings suggest that the flowers of OFI should be considered sources of antioxidants. The results of the present study revealed that OFI flower extracts have anticoccidial activities against *Eimeria*-spp.-induced infection in broiler chickens.

## 1. Introduction

*Opuntia ficus-indica* (OFI) L., known as the prickly pear spineless cactus, is a member of the family Cactaceae. This cactus is now widely distributed in arid and semi-arid regions, especially in the Mediterranean basin. Cactus is an important forage resource that serves as a source of water and energy for animals, mainly during the dry season [1]. Also, it produces edible, nutritionally rich fruits of excellent quality, and its cladodes serve as nutritious vegetables and salad dishes. In a scientific review, it is reported that nutritional value varies not only with environmental factors, mainly the soil and climate, but also according to the cactus species, cultivation location, and chemical characteristics of the soil [2].

Cactus flowers are large and have a beautiful yellow color, but they have no aroma [3]. It is reported that OFI flower extracts possess bioactive products that have potential use in the food industry, human and animal health, and cosmetic fields [4,5]. *Opuntia* flowers have long been used in folk medicine for a wide variety of therapeutic effects against ocular illnesses [6], diabetes [7], ulcers [8], and renal lithiases [9]. Moreover, the organic extracts of OFI flowers were studied for their antiacetylcholinesterase [10] and anti-inflammatory effects [11]. Furthermore, some *Opuntia* species showed inhibitory activity against *P. aeruginosa*, *S. aureus*, and *E. coli*, and others could be used as food preservative agents [12].

Several plant extracts that possess remarkable natural antioxidants and free radical scavenging potential may be useful to treat many diseases [13]. In addition, it is worth noting that plant compounds derived from secondary metabolism play a fundamental role in oxidative stress. These antioxidant molecules can donate hydrogen or electrons and are stable radical intermediates [14]. According to the literature, cactus pear can be used as a potential source of natural antioxidants in various applications due to its high nutritional value and bioactive phytochemicals [10]. Indeed, many reports have demonstrated the in vitro antioxidant activity of polyphenols extracted from the fruits, cladodes, flowers, and seeds of *Opuntia* spp. [15,16,17]. However, there are a few in vitro studies on the characterization of bioactive compounds and the potential antioxidant activity of OFI flowers, especially those grown in an Algerian bioclimate.

Avian coccidiosis is a complex intestinal disease caused by *Eimeria* spp., generating significant economic losses [18]. The resistance of the *Eimeria* species to anticoccidial drugs is one of the major problems in poultry livestock, with huge residual effects of anti-*Eimeria* drugs in meat. In recent years, special attention has been devoted to the use of plant extracts as an alternative treatment for coccidiosis in broiler chickens. It is reported that natural plant extracts are a promising choice to destroy and stop the replication of *Eimeria* oocysts [19]. Indeed, several scientific papers have been published on plant extracts and their phytochemicals for avian coccidiosis [20,21,22,23]. However, to the best of our knowledge, the anticoccidial activity of the *Opuntia ficus-indica* flower extract has never been reported.

The objective of the present study is to identify the biochemical compounds extracted from OFI flowers growing in Algeria using ultra-high-performance liquid chromatography–electrospray ionization quadrupole time-of-flight mass spectrometry (UPLC-ESI-Q-TOF-MS) and to evaluate their in vitro antioxidant activities. In addition, the in vitro anticoccidial effects of OFI flower extracts on the destruction of *Eimeria* oocysts isolated from naturally infected chickens have been investigated.

## 2. Materials and Methods

### 2.1. Plant Material

OFI flowers were collected after the post-flowering stage in May 2019 in Talendjast (36.6° N, 5.1° E), department of Bejaia, North of Algeria. These flowers were sorted and cleaned with a brush to eliminate spines and dust, ground, and then sieved.

### 2.2. Chemical Reagents

Folin–Ciocalteu and aluminium chloride were from Biochem, Chemopharma (Montreal, QC, Canada), sodium carbonate was from Biochem, Chemopharma (Cosne-Cours-sur-Loire, France), gallic acid was from Biochem-chemopharma (Cosne-Cours-sur-Loire, France), and all other chemicals were from Sigma Chemical (Sigma-Aldrich GmbH, Schnelldorf, Germany). Diclazuril was also used (Diclosol^®^, Avico, Arab Industry Veterinary Co., Amman, Jordan).

### 2.3. Extraction

A domestic microwave (NN-S674MF, Maxi Power Enterprise Ltd., Shenzhen, Guangdong, China) was used with a refrigerant to condense the vapors generated during the extraction [24]. Briefly, 1 g of the homogenous OFI flower powder was mixed with acetone. After a pre-leaching time (5 min), the suspension was irradiated by using the microwave under different experimental conditions. The suspension was cooled in ice water immediately after the extraction. Then, the OFI extract was centrifuged at 4500 rpm for 10 min (Nuve, NF-800-R Model, Ankara, Turkey) and filtered.

### 2.4. Determination of Phenolic Compounds

#### 2.4.1. Determination of Total Phenolic Content (TPC)

The content of OFI flowers’ phenolic compounds (TPC) was determined according to Velioglu et al. [25]. Briefly, 1.5 mL of diluted Folin–Ciocalteu reagent and 0.5 mL of OFI extract were mixed. Then, 1.5 mL of sodium carbonate (6%) was added after 5 min. The reaction mixture was incubated for 1 h in complete darkness and the absorbance was measured at 750 nm compared to a blank (Uvline 9400, Secomam, Alès, France). The calibration curve’s standard was gallic acid. The outcomes are presented as the gallic acid equivalent (GAE) in mg per g of dry weight (DW).

#### 2.4.2. Determination of Flavonoid Content

The flavonoid content in the OFI flower extracts was estimated according to the method described by Bahorun et al. [26]. The AlCl3 reagent (2%) and OFI extract were combined in equal parts. After 30 min of incubation, the absorbance of the mixture reaction was measured at 420 nm wavelength. The measured total flavonoid contents were expressed as mg quercetin equivalents (QE) per g dry weight (DW).

### 2.5. Antioxidant Activities

#### 2.5.1. DPPH Radical Scavenging Activity

The radical scavenger 1,1-diphenyl-2-picrylhydrazyl (DPPH) was used to assess the antioxidant activity of the OFI flower extract according to the method described previously by the authors of [27]. An aliquot (200 µL) of the extract was mixed with 1 mL of a 60 M methanolic DPPH solution. After 30 min, the decolorizing process was recorded at a wavelength of 515 nm. The amount of antioxidant power per gram of dry weight was represented as milligrams of gallic acid equivalent (GAE) (DW).

#### 2.5.2. Hydrogen Peroxide Scavenging Activity

The scavenging capacity of the OFI flower extracts for hydrogen peroxide was determined according to the method described by Ruch et al. [28]. One hundred and fifty microliters of OFI flower extracts were mixed with 1 mL of H_2_O_2_ (40 mM) in phosphate buffer, and another 1350 µL of phosphate buffer solution (0.1 mM, pH 7.4). After 10 min of incubation, the absorbance was measured at a wavelength of 230 nm. The measured hydrogen peroxide scavenging activity was expressed as equivalent mg gallic acid (GAE) per g of dry weight (DW).

### 2.6. Phenolic Compound Profile

The optimum phenolic compound profile of the OFI flower was obtained by using ultrahigh-performance liquid chromatography–electrospray ionization quadrupole time-of-flight mass spectrometry (UPLC-ESI-Q-TOF-MS) as previously performed by the authors of [24].

### 2.7. Evaluation of the Anticoccidial Activity

#### 2.7.1. *Eimeria* Oocysts Isolation and Purification

In the current study, the used *Eimeria* spp. oocysts were isolated from the fresh feces of broilers suffering from coccidiosis in the Bejaia area (Algeria). The oocysts were sporulated during incubation in 2.5% K_2_Cr_2_O_7_ solution in the presence of a suitable humidity (>70%) and temperature (29 °C) [29]. Subsequently, the sporulated oocysts were washed and counted using a Malassez chamber. The mean number of oocysts per milliliter of the sample was calculated. The identification of *Eimeria* species in chickens was based on a previously described standard parasitological technique [29]. The oocysts were identified according to their shape, the presence or absence of the micropyle, their sporulation time, intestinal location, and appearance, as well as according to the coarse characteristics of the intestinal lesions. The percentage of each identified species in the mixed suspension was as follows: *Eimeria tenella* 75%, *Eimeria acervulina* 10%, *Eimeria mitis* 4%, *Eimeria preacox* 4%, *Eimeria maxima* 4%, and *Eimeria brunetti* 3%.

The purification of the oocysts was carried out from one liter phosphate-buffered saline (PBS, containing 8 g/L NaCl, 0.2 g/L KCl, 1.13 g/L Na_2_HPO_4_, 2H_2_O, and 0.2 g/L KH_2_PO_4_) with some modifications. Neutral substrates, containing antibiotics (penicillin V 100 IU), were added to prevent bacterial growth, and fluconazole (17 mg/mL) was added as an antifungal agent. The pH was adjusted to 7.4 and the solution was sterilized by membrane filtration using a 0.2 μm filter. The HBSS (Hanks’ Balanced Salt Solution) medium was prepared in the laboratory (NaCI, 8.0; KCl 0,4; CaCl_2_, 0.139; D-glucose, 1.0; Na_2_HPO_4_, 0.0478; KH_2_PO_4_, 0.06; and MgSO_4_, 0.097 g/Lin one-liter distilled water). The solution was sterilized, as was that of the 0.2% agar.

#### 2.7.2. Effects of OFI Flower Extract on the Decrease in Oocysts Number

The anticoccidial activity of the OFI flower extract was determined in triplicate by incubation at ambient temperature for 24 h. The suspension solution was incubated for, 1, 3, 5, 7, and 24 h. One milliliter suspension contained 100 µL of washed suspension of *Eimeria* oocysts at a concentration of 24.5 × 10^5^ oocysts/mL, 700 μL of PBS, and another 200 μL of the optimum OFI flower extract. After incubation, the samples were centrifuged at 320 g for 5 min, and the absorbance of the supernatant was measured at a wavelength of 273 nm using a spectrophotometer (Shimadzu, model: UV 100, Kyoto, Japan). The percentage of the destroyed and sporulated oocysts was then calculated using the following equation:$$N_{r}=100-(\frac{N_{t}\times 100}{N_{0}})$$
where *N_r_* is the reduction rate of the number of; *N_t_* is the number of oocysts at x time (1, 3, 5, 7, and 24 h); and *N*_0_ is the number of oocysts at time 0 (24.5 × 10^5^ oocysts).

As a positive control, diclazuril (Algicox 10 mg/mL, anticoccidial) was tested at a concentration of 10 mg/mL. An acetone solvent was also used as the negative control. The lethal concentration LC_50_ of both the OFI flower and diclazuril values was then inferred from the regression curve.

### 2.8. Statistical Analyses

In the present study, a three-level, four-factorial Box–Behnken experimental design (BBD) from the JMP program (Version 10.0, SAS package, Heidelberg, Germany) was applied to validate the extraction process parameters affecting the rate of total phenolic content (TPC), flavonoid, DPPH radical scavenging, and hydrogen peroxide scavenging activity (H_2_O_2_) of OFI flower extract. The values of the factors were coded as three levels in BBD as follows: −1 (low), 0 (middle point), and 1 (high). Four factors were selected: acetone concentration (X_1_, 40–80%), liquid-to-sample ratio (X_2_, 30–50 mL/g), extraction time (X_3_, 30–150 s), and microwave power (X_4_, 400–1000 Watts). All data analyses were performed using the Statistica 12.0 software. Experiments were conducted in triplicate, and the results were presented as mean ± standard deviation (SD).

## 3. Results and Discussion

### 3.1. Bioactive Compounds

Due to the growing interest in the health benefits of plant constituents, it is essential to evaluate and optimize the parameters of the plant extraction process. In this study, we investigated the effects of extraction variables on the phenolic content and antioxidant activities of OFI flowers and identified their optimum combinations. As shown in Table 1 and Table 2, a Box–Behnken design was applied to optimize the microwave parameters (acetone concentration, ratio, irradiation time, and microwave power) to maximize the extraction of phenolic and antioxidant compounds from OFI flowers. The prickly pear flower TPC ranged from11.48 to 34.66 mg GAE/g DW, the DPPH radical varied from 31.39 to 41.32 mg GAE/g DW, the flavonoid varied from 08.59 to 25.71 mg EQ/g DW, and H_2_O_2_ scavenging activity varied from 5.24 to 8.19 mg GAE/g DW, respectively. These results highlight the considerable influence of the optimized parameters mentioned above on the phenolic content and antioxidant activities. It has been previously reported that the extraction process parameters, such as the sonication temperature and time, the solvent type, and polarity, have a considerable influence on the release of phenolic compounds from the solid matrix and can influence the antioxidant activities of the obtained extract [30,31]. Moreover, numerous studies have demonstrated that the solvent concentration, solvent-to-sample, temperature, and pressure rate parameters affect the extraction yield [32,33,34]. Our results are in agreement with those of Bansodet al. [35] and Tsiaka et al., [36], who reported that the optimization parameters influence the extraction of both bioactive compounds and antioxidant activities. In the current study, special interest has been given to microwave-assisted extraction (MAE) due to its positive impact on the extraction of bioactive compounds, i.e., higher product yields and shorter extraction times [37,38].

According to the data presented in Table 3, the F values of the four models, namely TPC, flavonoid, DPPH radical, and H_2_O_2_ scavenging activities, were significantly higher (*p* < 0.0001 variations in response, which would probably be due to factor effects). The lack of fit for each model was not significant (*p* > 0.05). The determination coefficients (R^2^) of TPC, flavonoid, and DPPH radical activity models were 0.94, and the values of the adjusted determination coefficients (R^2^ adj.) ranged from 0.87 to 0.88, whereas R^2^ and R^2^ adjusted for the H_2_O_2_ activity were 0.88 and 0.74, respectively. The experimental analyses of the TPC, flavonoid, DPPH radical, and H_2_O_2_ scavenging activities of the OFI flower extract were conducted for each response using the optimized extraction conditions. The obtained results were compared with the predicted values. Our results revealed significant differences between the predicted and experimental values of the models. Figure 1 illustrates the ideal parameters for maximizing the microwave-assisted extraction of bioactive compounds and antioxidant activities of the OFI flower, namely acetone content (73.59%), ratio (47.20 mL/g), extraction time (30 s), and power (1000 W). The response values of TPC, flavonoids, DPPH radical, and H_2_O_2_ scavenging activities were 35.15, 23.45, 39.10, and 8.57 mg standard/g DW, respectively. The optimal extract of OFI flowers contains 36.71 ± 0.12 mg GAE/g DW for total polyphenols and 24.85 ± 0.43 mg QE/g DW for flavonoids concentration. On the other hand, the results of the DPPH radical and H_2_O_2_ scavenging activity were 40.05 ± 0.20 and 8.54 ± 0.08 mg GAE/g DW, respectively. In the present study, the total phenolic compounds and flavonoid contents of the OFI flower extract were significantly higher than those of the OFI flowers grown in Tunisia [4], which were 7.02 mg GAE/g DW and 2.05 mg RE/g DW, respectively. Moreover, Berrabah et al. [39] reported low values of total phenolic (10.89 ± 5.60 mg/g GAE) and flavonoid (0.96 ± 0.33 mg/g QE) contents of the methanolic extracts of OFI flowers compared with our findings. In a recent study, Brahmi et al. [17] optimized the extractions of the OFI flower phenolic compound using three different methods, namely maceration, soxhlet, and ultrasound-assisted extraction. The obtained results highlighted that the total phenolic compounds optimized by three methods were lower (5.66 ± 0.37, 1.96 ± 0.59, and 24.38 ± 0.82 mg GAE/g DW, respectively) than those observed in the present study. Likewise, the flavonoid contents varied from 1.05 ± 0.01 to 9.70 ± 0.10 QE/g DW. Furthermore, the extraction time varied from 1 to 24 h, depending on the extraction method, whereas it was 30 s in the present study. These results demonstrate that microwave-assisted extraction is more efficient for the extraction of bioactive compounds.

The identification of the phenolic compounds in the OFI flower extract using ultra-performance liquid chromatography coupled with time-of-flight mass spectrometry (UPLC-ESI-Q-TOF-MS) revealed the presence of twelve phenolic molecules (Table 4), including phenolic acids, flavonoids, and their derivatives (quinic acid, coumaric acid, piscidic acid, eucomic acid, isorhamnetin, kaempferol-3-O-rutinoside, kaempferol 3-O-arabinoside, quercetin, isoquercetin, isorhamnetin 3-O-rutinoside, isorhamnetin 3-O-glucoside, and quercetin-3-O-rutinoside). Indeed, De Leo et al. [40] identified kaempferol 3-O-arabinoside, isorhamnetin 3-O-glucoside, and quercetin 3-O-rutinoside in OFI flower methanol extracts from Italy. Moreover, many studies have reported the presence of phenolic acids, such as coumaric acid, quinic acid, piscidic acid, eucomic acid, and other compounds in fruit, pulp, and peel OFI extracts [17,31,41,42]. Also, Amrane-Abider et al. [10] identified some phenol compounds, namely quercetin-3-O-rutinoside, isorhamnetin, isoquercetin, and kaempferol-3-O-rutinoside in *Opuntia ficus-indica* peel and flower teas using the boiling water method. These differences may be due to genetic factors, prickly pear varieties, culture or growth conditions, and geographical variations in prickly pear plants [43]. In addition, the extraction method can alter the total phenolic content [44]. Also, variations in phenol compounds between the studies can be related to the stage of OFI flower maturity.

### 3.2. Antioxidant Activities

Reactive oxygen species (ROS) are small molecules derived from oxygen molecules, including free oxygen radicals, such as superoxide and hydrogen peroxide, which interact with transition metal ions. These reactive hydroxyl radicals can have harmful effects on health [45,46]. In recent years, the demand for natural antioxidants, especially those of plant origin, has considerably increased due to the potential toxicological effects of synthetic antioxidants [47,48]. Natural antioxidants from plant extracts have been investigated for their ability to reduce inhibition by scavenging initiating radicals, breaking chain reactions, decomposing peroxides, decreasing localized oxygen concentrations, and binding with chain-initiating catalysts. The antioxidant properties of plant extracts can be evaluated by DPPH radical scavenging activity, hydrogen peroxide scavenging activity, and others. In the present study, the results of the DPPH radical test were higher than those previously reported by several authors. Brahmi et al. [17] reported that DPPH radical values of the OFI flower optimum varied from 13.76 ± 0.27 to 34.78 ± 0.87 μmol Trolox Equivalent/g DW, which corresponded to 3.44–8.7 mg GAE/g DW. In addition, our results were higher than those of OFI seed extracted by microwave-assisted extraction, (2.30 ± 0.27 mg GAE/g DW) as noted by Amrane-Abider et al. [24]. Regarding hydrogen peroxide scavenging, the results of OFI flower extracts exhibited strong activity of 8.54 ± 0.08 mg GAE/g DW, which corresponded to 96%. The data from this study were consistent with those published previously, which reported that the hydrogen peroxide scavenging activity of OFI fruits ranged from 76 to 96% [49]. However, the hydrogen peroxide scavenging activity of the OFI flower extract was slightly higher than that of the OFI seed extract (91.87–93.55%) [50]. These differences in antioxidant activity may be affected by the extraction method and the solvent used, which strongly influence the extract composition [51].

### 3.3. Anticoccidial Activity

Poultry livestock are seriously affected by coccidiosis; thus, exploration of other alternative natural sources is necessary to reduce the treatment cost and the resistance against *Eimeria* parasite. The use of plant extracts as alternative prophylactic or therapeutic strategies could effectively reduce coccidiosis in poultry farms [52]. The present study was carried out in order to investigate the in vitro effect of OFI flower extract on *Eimeria* spp. collected from naturally infected broiler chickens. As shown in Figure 2, the lethal concentrations (LC_50_) of OFI extract optimum and diclazuril were 66.04 mg/mL and 32.03 mg/mL, respectively. The in vitro effect of the OFI flower extracts on *Eimeria* oocysts after 7 h of treatment recorded a decreasing rate of 44.89%. However, the anticoccidial effects of diclazuril resulted in a strong decrease in the oocyst number, i.e., 54.69%. As has been reported in previous studies, diclazuril had significantly higher (*p* < 0.05) anticoccidial activity against coccidia oocysts after 24 h [53] and 48 h of incubation [54]. The results of this study showed a correlation between diclazuril (R^2^ = 0.82) and OFI flower extract (R^2^ = 0.91) concentrations in reducing the oocyst numbers (Figure 3). Biological compounds, such as polyphenols, tocopherols, and flavonoids, have demonstrated strong in vitro and in vivo anticoccidial properties [52,55,56]. Furthermore, Debbou-Iouknane et al. [22,23] showed that leaf olive extracts rich in natural polyphenolic components, such as quercetin and oleuropein, possess the ability to destroy *Eimeria* spp. It is also important to mention that plant constituents can have many beneficial individual or combined properties that are rich in bioactive elements.

In this study, OFI flower extract inhibited sporulation and damaged the morphology of *Eimeria* oocysts compared with normal sporulated *Eimeria* oocysts containing sporocysts. The destructive effect is the result of intracellular content release, such as aromatic amino acids and nucleotides, which are expressed by UV-absorbing substances after treatment [57]. Similar observations have been reported by Abbas et al. [58], who revealed that *Vitis vinifera* extract showed in vitro anticoccidial effects by affecting sporulation and damaging the morphology of *Eimeria* oocysts in chickens. In another study, the extracts of *P. emblica* remarkably inhibited oocyst sporulation, reduced the oocyst infectivity, lowered the fecal oocyst excretion, and reduced the pathogenicity of *E. tenella* in chickens [59]. This anticoccidial effect is probably due to the abundance of polyphenols in the OFI flower extract. Moreover, it has been reported that the plant extracts could penetrate both layers of the oocyst shell and cause a loss of intracellular components, resulting in the destruction and softening of the central cytoplasmic mass of the parasite [58]. 

Our results are in agreement with those of previous investigations, showing the efficacy of plant extracts against mixed or individual *Eimeria* infections, which can be used as potential anticoccidials in broiler chickens [60]. In addition, another study suggested the use of a combination of plant extract multi-compounds induces a good response, as an immunostimulatory effect, to destroy the oocyst of *Eimeria* or interfere with their life cycle [61]. It has also been reported in the literature that phenolic components demonstrate a wide range of biological properties, such as antibacterial, anticancer, antiproliferative, and anti-inflammatory properties [62,63,64]. In another research, Zaman et al. [65] demonstrated in vivo anticoccidial effects of different concentrations of a complex based on four plants, including *Plantsirachta indica* and *Nicotiana tabacum* leaves, *Calotropis procera* flowers, and *Trachyspermum ammi* seeds in broiler chickens in comparison with the amprolium standard anticoccidial drug. Moreover, the secondary metabolites of plant extracts have been used for their antiprotozoal activities, especially against *Plasmodium*, *Leishmania,* and *Trypanosoma* spp. [66]. In the present study, the destruction of *Eimeria* oocysts could be attributed to the constituent bioactivity of the OFI flower extract and the nature of the compounds in the functional group [67].

Among the main bioactive components of the OFI flower extract is quercetin. It is known that quercetin inhibits the synthesis of HSP90, HSP70, and HSP27 in *Toxoplasma gondii*, which are described as virulence factors [68]. These alterations promote reductions in the invasion of the host tissues, adhesion, proliferation, and cell differentiation [69,70]. In other studies, it has been reported that kaempferol promotes the inhibition of pyruvate kinase, dihydroorotase enzyme (LdDHOase), and cytidine deaminase activities of *Leishmania donovani*, which impact the pyrimidine biosynthesis pathway that causes parasite death [71,72]. The anti-*Eimeria* effect could be explained by the fact that biophenol molecules rapidly diffuse through the cell membrane parasite because of their polarity.

## 4. Conclusions

The optimization of the extraction procedure for phenolic compounds and antioxidant activities of the *Opuntia ficus-indica* flower extract was successfully examined using a Box–Behnken response surface design. The optimized conditions were validated and were found to fit very well with the experimental values. The extraction of OFI flower phenolics highly depends on the solvent acetone concentration, ratio, irradiation time, and microwave power. OFI flowers have been found to be a source of natural phenolic compounds, such as eucomic acid, quercetin, isorhamnetin 3-O-rutinoside, and quercetin-3-O-rutinoside. These findings suggest that OFI flowers should be considered a source of antioxidants. To our knowledge, this is the first research on the effect of OFI flower extract on *Eimeria* oocysts in broiler chickens. The results of the present study revealed that OFI flower extracts showed anticoccidial activity against *Eimeria* spp.-induced infections in broiler chickens. This could be a promising alternative for the synthesis of drugs to control coccidiosis in poultry. However, further studies should be carried out to test the in vivo efficiency of the OFI flower bioactive compounds in broiler chickens.

## Figures and Tables

**Figure 1 biomedicines-11-02173-f001:**
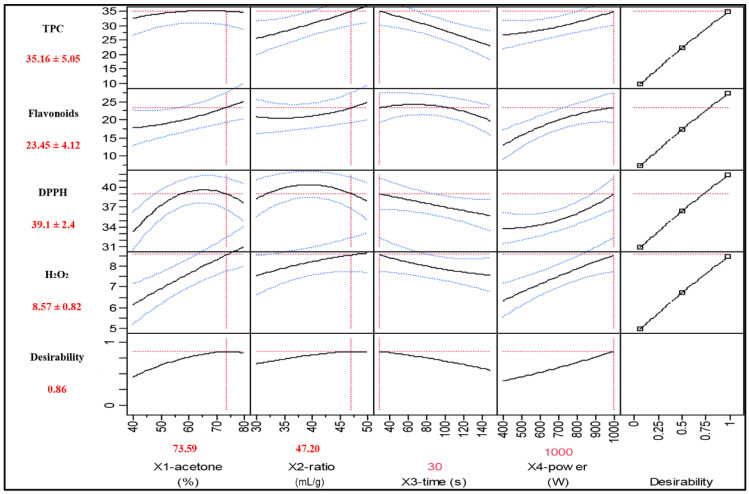
The ideal parameters to maximize the microwave-assisted extraction of bioactive compounds and antioxidant activities of OFI flower. Blue color represents the confidence interval; red line represents the optimun in the surface profile.

**Figure 2 biomedicines-11-02173-f002:**
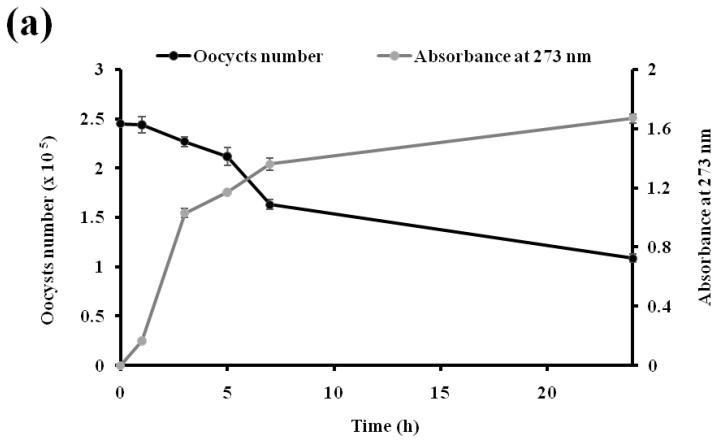
Kinetics of the oocysts number and material release from *Eimeria* oocysts treated by optimum of *Opuntia ficus*-indica flower extract (**a**) and diclazuril (**b**).

**Figure 3 biomedicines-11-02173-f003:**
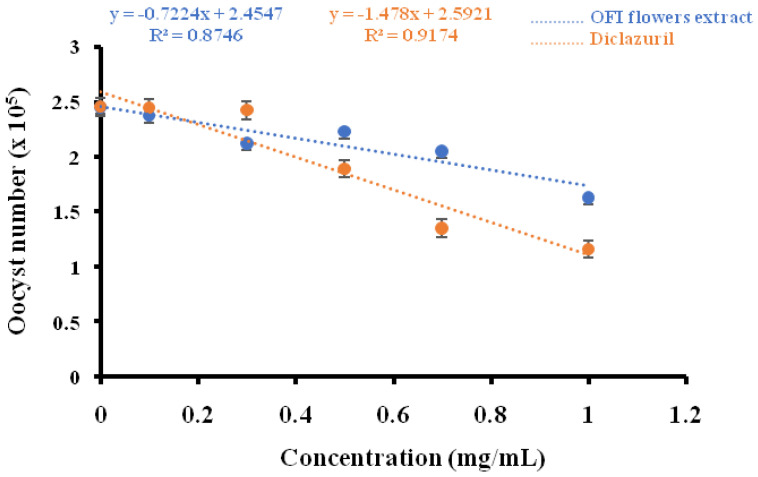
The correlation between the optimum of *Opuntia ficus-indica* flower extract and diclazuril concentrations, and the number of *Eimeria* oocysts.

**Table 1 biomedicines-11-02173-t001:** Range of coded and actual values for Box–Behnken design.

Factor	Level		
	−1	0	+1
*X* _1_	40	60	80
*X* _2_	30	40	50
*X* _3_	30	90	150
*X* _4_	400	700	1000

Legend: *X*_1_, acetone concentration (% *v*/*v*, solvent/water); *X*_2_, ratio (mL/g); *X*_3_, time (seconds); *X*_4_, power (Watt).

**Table 2 biomedicines-11-02173-t002:** Box–Behnken experimental design of extraction parameters for the optimization of TPC, flavonoids, DPPH, and H_2_O_2_ scavenging activities of *Opuntia ficus-indica* flower extracts obtained using microwave.

Run		TPC (mg GAE/g DW)		Flavonoid (mg QE/g DW)		DPPH (mg GAE/g DW)		H_2_O_2_ (mg GAE/g DW)	
	Pattern *X*_1_ *X*_2_ *X*_3_ *X*_4_	Observed	Predicted	Observed	Predicted	Observed	Predicted	Observed	Predicted
1	+ 0 + 0	23.854	22.172	20.212	20.292	32.606	33.429	7.225	7.342
2	− 0 + 0	11.592	10.086	08.598	08.262	34.574	34.559	7.637	7.754
3	00 + +	13.039	15.362	11.075	09.371	37.321	37.524	7.327	6.919
4	+ 00 −	23.322	24.292	17.907	17.136	33.254	32.809	6.636	6.495
5	+ + 00	25.780	25.344	25.716	25.896	32.056	31.997	7.924	8.115
6	+ 0 − 0	24.165	26.318	20.013	19.342	33.693	33.774	7.543	7.193
7	− − 00	11.488	10.704	10.545	11.920	31.398	31.128	7.184	7.209
8	0 − + 0	14.938	15.450	10.518	10.640	34.782	34.600	6.833	6.991
9	− + 00	25.787	26.722	19.194	20.823	31.874	32.551	6.183	6.375
10	− 00 +	16.054	14.658	09.935	10.160	32.610	33.318	6.402	6.560
11	+ − 00	25.834	24.679	22.079	22.004	31.452	30.445	5.870	5.895
12	0+ − 0	34.663	33.725	21.207	20.539	35.475	35.921	7.053	6.911
13	0−0−	24.057	24.523	15.136	14.442	34.352	35.389	6.630	6.280
14	0000	21.092	22.979	16.447	17.316	36.282	37.780	6.993	7.126
15	0 − − 0	22.289	22.399	17.336	16.288	34.013	34.879	6.366	6.558
16	0 + 0 −	28.416	28.211	20.858	19.531	37.904	38.414	7.453	7.270
17	− 0 0 −	24.020	24.443	17.371	15.567	38.191	37.848	7.277	7.135
18	0 0 − +	30.280	29.170	17.326	17.542	40.922	40.819	7.688	7.713
19	0 + 0 +	29.349	29.530	21.755	21.442	37.737	36.766	7.135	7.252
20	0000	25.964	22.979	18.664	17.316	38.529	37.780	7.293	7.126
21	0 0− −	31.172	28.629	09.011	12.270	37.999	37.467	5.242	5.867
22	+ 00 +	28.256	27.407	22.492	23.749	36.512	37.119	7.468	7.626
23	0 ++ 0	21.344	20.808	18.683	19.184	37.135	36.533	8.199	8.024
24	0000	21.881	22.979	16.838	17.316	38.529	37.780	7.093	7.126
25	00 + −	21.683	22.572	12.099	13.437	41.322	41.096	8.017	8.208
26	0 − 0 +	15.684	16.535	13.416	13.736	37.261	36.817	6.905	6.855
27	− 0 − 0	23.478	25.806	17.303	16.215	34.639	33.881	6.706	6.356

Legend: *X*_1_, acetone concentration (%*v*/*v*, solvent/water); *X*_2_, ratio (mL/g); *X*_3_, time (seconds); *X*_4_, power (Watt).

**Table 3 biomedicines-11-02173-t003:** Analysis of variance (ANOVA) for TPC and DPPH values of TPC, flavonoids, DPPH, and H_2_O_2_ scavenging activities of *Opuntia ficus-indica* flower extracts.

Model	TPC	Flavonoid	DPPH	H_2_O_2_
DF	14	14	14	14
SS	871.601	525.810	193.409	10.070
F value	14.340	13.167	14.336	6.333
*p* value	<0.0001 *	<0.0001 *	<0.0001 *	<0.001 *
Lack of fit				
DF	10	10	10	10
SS	38.422	41.427	8.197	1.316
F value	0.5619	2.244	0.487	5.641
*p* value	>0.782	>0.347	>0.821	0.160
Pure error				
DF	2	2	2	2
SS	13.676	2.800	3.366	0.046
R^2^	0.94	0.94	0.94	0.880
R^2^_Adj_	0.88	0.87	0.88	0.740

Legend: DF, degree of freedom; SS, sum of squares; *, significantly higher differences.

**Table 4 biomedicines-11-02173-t004:** Phenolic compounds’ constituents identified in *Opuntia ficus-indica* flower extracts using microwave.

	Compounds	[M-H]	Retention Time (min)	Concentration (±SD, µg/g)	Molecular Formula	Fragment Ions (*m*/*z*)
1	Quinic acid	191	0.80	109.30 ± 0.39	C_7_H_12_O_6_	173 (87), 129 (69) 134 (58), 174 (28)
2	Coumaricacid	163	2.24	58.06 ± 0.17	C_9_H_8_O_3_	119 (100), 163 (22), 91 (2)
3	Piscidic acid	255	2.78	62.65 ± 0.10	C_11_H_11_O_7_	193 (100), 165 (48), 135 (30), 119 (18), 107 (3)
4	Eucomic acid	239	3.07	289.25 ± 0.21	C_11_H_12_O_6_	131 (75), 103 (5), 72 (1)
5	Isorhamnetin	315	3.91	80.68 ± 0.10	C_16_H_12_O_7_	315 (100), 301 (7), 297 (6), 285 (2)
6	Kaempferol-3-O-rutinoside	593	4.25	139.03 ± 0.05	C_27_H_30_O_15_	287 (100), 146 (7)
7	Kaempferol 3-O-arabinoside	417	5.45	89.98 ± 0.17	C_20_H_18_O_10_	285 (100), 227 (20), 417 (60)
8	Quercetin	301	6.79	258.16 ± 0.01	C_15_H_10_O_7_	273 (13), 179 (62), 151 (95), 155 (6), 121 (26)
9	Isoquercetin	463	6.99	226.05 ± 0.05	C_21_H_20_O_12_	301 (100), 300 (42), 179 (77)
10	Isorhamnetin 3-O-rutinoside	623	7.35	295.14 ± 0.01	C_28_H_32_O_16_	315 (42), 314 (100)
11	Isorhamnetin 3-O-glucoside	477	7.63	213.45 ± 0.11	C_22_H_22_O_12_	315 (100), 300 (80)
12	Quercetin-3-O-rutinoside	609	8.11	265.41 ± 0.04	C_27_H_30_O_16_	303 (100), 300 (55), 151 (12)

## Data Availability

All data were included in the present manuscript.

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
