# Peer review of "Bioactive Compounds and In Vitro Antioxidant and Anticoccidial Activities of Opuntia ficus-indica Flower Extracts"

_biomedicines, 2023, doi:10.3390/biomedicines11082173_

Round 1

Reviewer 1 Report

The article needs to be revised very carefully, namely eliminating duplicated words, separating words together (e.g. againstmixed) and correcting some designations, for example hour should be presented with a small h and not H as it is in the manuscript.

In Figure 1, the values shown on the yy and xx axis for the different parameters have 5 decimal places! My question is whether it is possible for these beings to have a quantification certainty that allows their values to be determined with this accuracy?

Line 296-297

The following phrase:

Reactive oxygen species (ROS) are small molecules derived from oxygen molecules including free oxygen radicals, such as superoxide and hydrogen peroxide, which are non-radicals, particularly upon interactions with transition metal ions.

Needs to be rephrased as the transition metals ions are here with any special importance!

In the legend of figures 2 and three it must be mentioned the conditions used to obtain the OFI extract optimum.

Figure 2 what it means Abserbency?

Figure 3a and b must be together on the same graph.

Author Response

Reviewer #1

Dear Reviewer,

Special thanks for your efforts in reviewing our manuscript and your valuable comments which have greatly contribute to increase its quality.

Comment 1: The article needs to be revised very carefully, namely eliminating duplicated words, separating words together (e.g. against mixed) and correcting some designations, for example hour should be presented with a small h and not H as it is in the manuscript.

Answer 1: Thank you for these pertinent remarks; the corrections have been made.

Comment 2: In Figure 1, the values shown on the yy and xx axis for the different parameters have 5 decimal places! My question is whether it is possible for these beings to have a quantification certainty that allows their values to be determined with this accuracy?

Answer 2: Experimental design and modeling are based on very precise algorithms, hence the importance of using them. We made the experimental design and checked some quality parameters and errors, such as lac of fit. At the end, we made the experiment with the experimental values proposed by the software (Figure 1), hence the experimental conditions were validated.

Comment 3: Line 296-297, The following phrase: “Reactive oxygen species (ROS) are small molecules derived from oxygen molecules including free oxygen radicals, such as superoxide and hydrogen peroxide, which are non-radicals, particularly upon interactions with transition metal ions. Needs to be rephrased as the transition metals ions are here with any special importance!

Answer 3: We have rephrased that paragraph (lines 296-299).

Comment 4: In the legend of figures 2 and three it must be mentioned the conditions used to obtain the “OFI extract optimum”. Figure 2 what it means “Abserbency”?

Answer 4: Lines 240–243 and Figure 1 depict the ideal extraction circumstances that let us reach the optimum. To avoid repetition, the authors suggest modifying the title of Figure 1 to make it clear that these are the optimal conditions.

Figure 1: The ideal parameters maximizing the microwave-assisted-extraction of bioactive compounds and antioxidant activities of OFI flower

Abserbency (or Absorbance) is correct. It means optical density, it is well explained in the section ’’Materials and Methods’’.

Comment 5: Figure 3a and b must be together on the same graph.

Answer 5: It has been corrected.

Thank you again!

Reviewer 2 Report

The fact that more studies should be carried out to test the in vivo efficiency of the OFI flower bioactive compounds in broiler chickens mean as this work is not completed at moment.  Hovewer the experimental values could be innovatives if validated 

Author Response

The fact that more studies should be carried out to test the in vivo efficiency of the OFI flower bioactive compounds in broiler chickens mean as this work is not completed at moment.  Hovewer the experimental values could be innovatives if validated.

Answer:

Dear Reviewer,

Thank you for your effortsin reviewing our mansucript and your comments to the quality of our submission. We hope that our efforts will be positively appreciated by the Editor, and the present version of the manuscript will be considered for publication in the prestigious Biomedicine journal. Moreover, addressing to the raised concerns by another reviewers in this review round have brought substantial improvements to the manuscript!

Thank you again!

Reviewer 3 Report

Thank you for allowing me the opportunity to review the paper titled "Bioactive compounds, in-vitro antioxidant and anticoccidial activities of Opuntiaficus-indica flowers extracts." The paper presents original research and provides valuable insights into the extracts of Opuntiaficus-indica flowers. The title, abstract, and introduction are well-written and effectively convey the context and importance of the study.

The "Materials and Methods" section is comprehensive, ensuring the study's reproducibility and offering a clear understanding of the techniques used to extract phenolic compounds from Opuntiaficus-indica flowers and validate their activities.

The alignment of the expected findings with the actual results adds credibility to the research. The study highlights the plant's potential as a source of antioxidant and anticoccidial effects, which could have significant implications in various applications.

The discussion section effectively interprets the results and provides a deeper understanding of their significance. Additionally, the conclusion provides a concise summary of the key points addressed in the study.

In my opinion, the paper's novelty and the way it presents the properties under investigation make a strong case for its acceptance in its current form. 

Author Response

Thank you for allowing me the opportunity to review the paper titled "Bioactive compounds, in-vitro antioxidant and anticoccidial activities of Opuntiaficus-indica flowers extracts." The paper presents original research and provides valuable insights into the extracts of Opuntiaficus-indica flowers. The title, abstract, and introduction are well-written and effectively convey the context and importance of the study.

The "Materials and Methods" section is comprehensive, ensuring the study's reproducibility and offering a clear understanding of the techniques used to extract phenolic compounds from Opuntiaficus-indica flowers and validate their activities.

The alignment of the expected findings with the actual results adds credibility to the research. The study highlights the plant's potential as a source of antioxidant and anticoccidial effects, which could have significant implications in various applications.

The discussion section effectively interprets the results and provides a deeper understanding of their significance. Additionally, the conclusion provides a concise summary of the key points addressed in the study.

Answer:

Dear reviewer,

Special thanks for your efforts in reviewing our manuscript and your valuable high class comments to the quality of our submission. We are delighted reading your words and we are very happy to be considered for publication in the prestigious Biomedicine journal!

Thank you again!

Round 2

Reviewer 1 Report

In figure 2 the legend in the graphic needs to be corrected for absorbance or optical density,  the word absorbency or Abserbency are not correct. The legend of figure 2 also needs to be correct.

Please correct the figure 2 legend for:

Figure 2. kinetics of the oocysts number and material release from Eimeria oocysts treated by optimum of Opuntia ficus-indica flowers extract (a) and diclazuril (b).

Author Response

Comment 1: In figure 2 the legend in the graphic needs to be corrected for absorbance or optical density,  the word absorbency or Abserbency are not correct.

Answer 1: Dear reviewer, once again thank you for your efforts and time for reviewing our manuscript. The requested change has been operated. Please see the revised version 

Comment 1: The legend of figure 2 also needs to be correct.

Please correct the figure 2 legend for:

Figure 2. kinetics of the oocysts number and material release from Eimeria oocysts treated by optimum of Opuntia ficus-indica flowers extract (a) and diclazuril (b)

Answer: The requested change has been operated.

Please see the revised version of the manuscript.

Thank you again!